# The Radiobiological Characterization of Human and Porcine Lens Cells Suggests the Importance of the ATM Kinase in Radiation-Induced Cataractogenesis

**DOI:** 10.3390/cells12162118

**Published:** 2023-08-21

**Authors:** Joëlle Al-Choboq, Thibaud Mathis, Juliette Restier-Verlet, Laurène Sonzogni, Laura El Nachef, Adeline Granzotto, Michel Bourguignon, Nicolas Foray

**Affiliations:** 1INSERM U1296 Unit “Radiation: Defense, Health, Environment”, 28 Rue Laennec, 69008 Lyon, France; joelle.al-choboq@inserm.fr (J.A.-C.); juliette.restier-verlet@inserm.fr (J.R.-V.); laurene.sonzogni@inserm.fr (L.S.); laura.el-nachef@inserm.fr (L.E.N.); adeline.granzotto@inserm.fr (A.G.); michel.bourguignon@inserm.fr (M.B.); 2Ophtalmology Department, Hospices Civils de Lyon, General University Hospital of Croix-Rousse, 103 Grande Rue Croix Rousse, 69004 Lyon, France; thibaud.mathis@chu-lyon.fr; 3MATEIS Laboratory, CNRS UMR5510, INSA, Université Claude-Bernard Lyon 1, Campus de la Doua, 69100 Villeurbanne, France; 4Department of Biophysics and Nuclear Medicine, Université Paris Saclay Versailles St Quentin-en-Yvelines, 78035 Versailles, France

**Keywords:** lens, cataractogenesis, ionizing radiation, ATM protein

## Abstract

Studies about radiation-induced human cataractogenesis are generally limited by (1) the poor number of epithelial lens cell lines available (likely because of the difficulties of cell sampling and amplification) and (2) the lack of reliable biomarkers of the radiation-induced aging process. We have developed a mechanistic model of the individual response to radiation based on the nucleoshuttling of the ATM protein (RIANS). Recently, in the frame of the RIANS model, we have shown that, to respond to permanent endo- and exogenous stress, the ATM protein progressively agglutinates around the nucleus attracted by overexpressed perinuclear ATM-substrate protein. As a result, perinuclear ATM crowns appear to be an interesting biomarker of aging. The radiobiological characterization of the two human epithelial lens cell lines available and the four porcine epithelial lens cell lines that we have established showed delayed RIANS. The BFSP2 protein, found specifically overexpressed around the lens cell nucleus and interacting with ATM, may be a specific ATM-substrate protein facilitating the formation of perinuclear ATM crowns in lens cells. The perinuclear ATM crowns were observed inasmuch as the number of culture passages is high. Interestingly, 2 Gy X-rays lead to the transient disappearance of the perinuclear ATM crowns. Altogether, our findings suggest a strong influence of the ATM protein in radiation-induced cataractogenesis.

## 1. Introduction

Only a few months after the discovery of X-rays, the first radiation-induced (RI) eye damage was observed in exposed laboratory animal models [1]. Later, during the first radiotherapy attempts, the first human RI ocular lesions were described by Chalupecky in 1897 [2]. The progressive application of radiotherapy, whether against cancerous or non-cancerous eye diseases, has led to a significant number of accidents [1]. In this context, one can consider that the first description of RI cataracts in humans was reported by Rollins in 1903 [3].

The lens is considered to be the most radiosensitive tissue among not only the ocular structures but also among all the organs [4,5]. However, it must be stressed that the term “radiosensitive” was originally used to describe RI toxicity rather than RI-accelerated aging [4,5]. Furthermore, one of the most current clinical consequences of exposure of the eye to ionizing radiation is the formation of cataracts, which cannot be considered an RI toxic effect [6,7]. Cataracts describe the clouding of the lens, reducing the amount of light reaching the retina [4,8,9]. Untreated, cataracts can lead to complete vision loss and are the most common cause of blindness. Cataracts represent 51% of cases of visual impairment worldwide [8,10]. Depending on the anatomical appearance and the location of the opacity in the lens, three major types of cataracts have been identified: Nuclear, cortical, and posterior or anterior subcapsular cataracts [4,5,10,11,12]. Epidemiological studies and animal experiments have shown that posterior subcapsular cataracts are the most common cases of RI cataracts [5]. Even though the first RI human cataract was observed in 1903, it was not until cataracts were observed in atomic bomb survivors that interest in radiation protection of the eye and lens started [13]. In 1954, the International Commission on Radiological Protection (ICRP) classified cataracts as a radiological health hazard and recommended a dose limit of 300 mrem/week for the lens [14]. Since 1969, cataracts have been considered tissue reactions with an associated dose threshold [13]. The last revision of the lens dose limit was proposed in 2011: The ICRP dramatically reduced the lens dose threshold from 2 to 0.5 Gy for acute exposures and the equivalent dose limit from 150 to 20 mSv/year (100 mSv in 5 years without any year exceeding 50 mSv) [15]. However, the ICRP report has not ruled out the possibility that a threshold does not exist [15]. It should be noted that the ICRP recommendations were primarily based on epidemiological studies, in particular those from atomic bomb survivors, radiotherapy patients, and medical workers.

To date, industrial and medical workers (notably interventional cardiologists, refs [16,17,18]) represent the largest cohorts of subjects likely to develop RI occupational cataracts [19]. However, among these exposed workers, the specific subset of astronauts and airline pilots are exposed to relatively low doses of radiation generated from mixtures of rays and particles such as gamma rays, low-energy protons, neutrons, and, to a lesser extent, heavy ions [19]. Hence, it is possible that RI cataractogenesis features and, notably, dose threshold may also strongly depend on the type of radiation [20]. Throughout its last report, the ICRP has therefore highlighted the lack of information on the mechanism of cataract development after exposure to low doses of ionizing radiation and the need for new mechanistic studies to better understand the process of RI cataractogenesis [15].

There are two major limits in the quest for the basic mechanisms of cataractogenesis. First, the lack of availability of epithelial lens cell lines, notably human ones, reflects not only the difficulties of cell sampling to establish cell lines in humans but also the difficulties of amplifying this type of cell up to high cell culture passages to mimic aging. To date, two human epithelial cell lines are available: The untransformed HLEpiC cell line from ScienCell (Carlsbad, CA, USA) and the transformed HLE-B3 cell line from ATCC (Manassas, VA, USA). It is noteworthy that, as documented with skin fibroblasts, the transformation or immortalization of cell lines may introduce significant biases in the radiation response, and therefore, any radiobiological data from the transformed HLE-B3 cell lines may be interpreted with caution.

Second, another difficulty in the cataractogenesis research area is the lack of reliable biomarkers for the RI aging process. Indeed, despite the considerable efforts provided by telomeres, telomerase activity, and residual DNA breaks, there is still no consensus about a molecular endpoint that would quantify a specific feature of aging [7].

A model based on the RI nucleoshuttling of the ATM kinase protein (RIANS) has been proposed to quantify the risk linked to exposure to ionizing radiation [21,22,23,24,25,26]. The RIANS model is based on the oxidative stress-induced monomerization of the ATM dimers, notably those that are very abundant in the cytoplasm. These RI ATM monomers diffuse in the nucleus, phosphorylate the H2AX histone variants (γH2AX) at the DNA double-strand break (DSB) sites, and form nuclear γH2AX foci [21,25]. These γH2AX foci are the early step of the recognition of the DSB repaired by the non-homologous end-joining (NHEJ) pathway, predominant in quiescent mammalian cells [21,25]. During DSB repair, two ATM monomers reassemble at the DSB sites, which form the trans-autophosphorylated nuclear ATM (pATM) foci [26]. Any delayed RIANS causes a radiosensitivity phenotype associated with high cancer risk or accelerated aging [7,26]. The RIANS model provides a reliable prediction of clinical radiosensitivity [21,22,23,24,25,26,27] and a unified molecular explanation of cellular radiosensitivity observed in numerous genetic syndromes [28,29,30,31,32,33]. Any delayed RIANS has been shown to be triggered by the abnormal cytoplasmic abundance of the mutated protein that causes the disease in the cytoplasm. These proteins, called “syndromic X-proteins” are phosphorylation substrates of ATM and form abundant ATM–X multiprotein complexes that prevent RIANS [28,29,30,31,32,33]. A delay in RIANS may also cause a delay in the ATM-dependent inhibition of the MRE11 nuclease activity, responsible for the generation of DNA breaks. The phosphorylation of MRE11 by ATM is associated with the production of nuclear MRE11 foci [32]. Interestingly, the radiosensitive cancer syndromes have been associated with early MRE11 foci, while the radiosensitive aging syndromes are characterized by late MRE11 foci [32]. Furthermore, more recently, the X-proteins associated with aging syndromes have been found either attached to the nuclear membranes or in close proximity to them. Consequently, in cells from aging syndromes, ATM monomers bound to the X-proteins may specifically form perinuclear pATM crowns, easily quantifiable by immunofluorescence [33,34]. Altogether, the RIANS model biomarkers may discriminate between cancer proneness and aging [33,34]. To verify and document such hypotheses, the goal of this study was to perform the radiobiological characterization of human and porcine epithelial lens cells by using RIANS biomarkers and to identify some X-proteins that would be specific to lens aging and the process of RI cataractogenesis.

## 2. Materials and Methods

### 2.1. Human Cell Lines

Two radioresistant skin fibroblast cell lines (1BR3 and MRC5) that originated from apparently healthy patients were used in this study as negative controls for radiosensitivity. These cell lines were purchased from the European collection of authenticated cell cultures (ECACC, UK Health Security Agency, Porton Down, Salisbury, UK) under the references, #90011801 and #05011802, respectively. Fibroblasts were routinely cultured as monolayers with Dulbecco’s Modified Eagle’s Minimum Medium (DMEM) (Gibco-Invitrogen-France, Cergy-Pontoise, France), supplemented with 20% fetal calf serum, penicillin, and streptomycin.

The untransformed human lens epithelial cell line (HELpiC; #6550) and the untransformed human retinal pigment epithelial cell line (HRPEpiC; #6540) were purchased from ScienCell Research Laboratories (Carlsbad, CA, USA). Note that Hamada’s group mentioned that ScienCell Research Laboratories proposed several lots of primary normal human lens epithelial cells derived from different donors (likely from fetal origin) under the same name (HLEpiC) [35]. Hence, great confusion can occur with regard to the specificity of the radiation response of the HLEpiC cell lines according to the status of the different donors. Epithelial lens cells were routinely cultured as monolayers with epithelium cell medium (EpiCM), supplemented with 2% fetal bovine serum (ScienCell) and 1% growth supplement (EpiCGS, ScienCell), rinsed with PBS 1X (without Ca^2+^ and Mg^2+^, ScienCell), and then trypsinized with a trypsin/EDTA solution (ScienCell). The immortalized human lens epithelial cell line (HLE-B3) was purchased from the American Type Culture Collection (ATCC, Manassas, VA, USA) under the #CRL-11421 reference and was routinely cultured in the same conditions as indicated for human fibroblasts.

### 2.2. Porcine Cell Lines

One radioresistant porcine skin fibroblast cell line (PSF-1) that originated from a cutaneous biopsy from a butcher pig from a local slaughterhouse was used as a negative control for radiosensitivity. This biopsy was performed in the 1990s at Institut Gustave-Roussy (Villejuif, France) in agreement with the regulations of the French departments and the slaughterhouse veterinarians.

Four epithelial lens porcine cell lines were used in this paper. Porcine eyes were purchased from a local slaughterhouse (Abattoir des Crets, Bourg-en-Bresse, France) in agreement with the local regulatory department and the slaughterhouse veterinarians. This experimental step was performed with the assistance of T.M., and the procedure was described elsewhere [36]. This procedure, performed under reference number 69388005, was consistent with the European initiative for restricting animal experimentation, as no animals have been killed for our experimentation. Porcine eyes were maintained for 2 or 3 h after enucleation in CO_2_-independent serum (Thermo Fisher Scientific, Waltham, MA, USA). The eyes were cleaned for muscles and immersed for a few minutes in an antiseptic solution (Pursept-A Xpress, Merz Hygiene GmbH, Ebersbach an der Fils, Germany). The anterior segment of the bulb was removed, as well as the lens, the vitreous, and the retina. Lenses were collected in a saline solution at room temperature. Lenses were then placed on Petri dishes with their anterior side down. A small incision was made on the posterior capsule using sterile scalpels. The capsules were then opened and fixed on the Petri dishes using sterile forceps. The fibrous cellular mass was discarded, and only the capsules were kept (Figure 1).

The capsules were incubated and cultured by following the same procedure as for fibroblasts, rinsed with PBS 1X (with Ca^2+^ and Mg^2+^), then trypsinized with TrypLE Express 1X solution (Gibco). All the experiments were performed with cells in the plateau phase of growth (95–99% in G0/G1). The cells used in the experiments were cultured at different passages (from passage 3 to passage 30).

### 2.3. X-rays Irradiation

Irradiations were performed with a 6 MeV photon medical irradiator (SL 15 Philips) (dose rate: 6 Gy·min^−1^) [21].

### 2.4. Immunofluorescence

The immunofluorescence protocol was described elsewhere, together with the references for the primary anti-*γH2AX*, -*pATM*, and -*MRE11* and secondary antibodies [21,28,29,30,31,32,33]. The polyclonal anti-rabbit anti-*BFSP2* antibody (#PA5-98405, Invitrogen, Waltham, MA, USA) and the polyclonal anti-rabbit anti-*FYCO1* antibody (bs-13237R, Bioss antibodies, Wobum, MA, USA) were used at 1:100. Nuclei were counterstained with 4′,6′-Diamidino-2-phenylindole (DAPI, Cliniscience, Nanterre, France) antibodies [21,28,29,30,31,32,33].

The average size of the nuclei of each cell line used in this study was measured during immunofluorescence experiments with DAPI-stained nuclei and assessed using an Olympus BX51 fluorescence microscope and the dedicated Olympus Cells.2 software approved by the manufacturer. Considered spheric, the nucleus surface S was calculated by the formula S = 4πR^2^ in which R was defined as the average nucleus radius (Table 1).

### 2.5. Micronuclei Assays

The DAPI counterstaining permitted the quantification of the micronuclei [28,29,30,31,32,33]. It must be stressed here that the micronuclei data assessed may not be numerically equal to those obtained with the micronucleus assay involving cytochalasin B [37,38]. 

### 2.6. Proximity Ligation Assay

The proximity ligation assay (PLA) allows the visualization of endogenous protein–protein interactions at the single-molecule level [39]. The PLA assay protocol is detailed elsewhere [31]. PLA foci were assessed with similar procedures to those described for nuclear foci (see Section 2.4).

### 2.7. Cell Extracts and Immunoblots

The procedures of the cell extracts and immunoblots are described elsewhere [28,29,30,31,32,33]. 

### 2.8. Statistical Analysis

A two-way ANOVA was used to compare numerical values, and Spearman’s test was used to compare the kinetic data. The foci kinetic data were fitted to the so-called Bodgi’s formula [40]. Statistical analysis was performed using Kaleidagraph v4 (Synergy Software, Reading, PA, USA) and GraphPad Prism (San Diego, CA, USA).

## 3. Results

### 3.1. Radiobiological Features of the Untransformed Human Lens Cell Lines

Unrepaired DNA breaks may propagate up to mitosis and lead to chromosome fragments that may escape from the metaphases, generate micronuclei, and cause mitotic death. A quantitative correlation was found between residual (24 h post-irradiation) micronuclei and cellular radiosensitivity [7,41]. Twenty-four hours after the irradiation, the radioresistant control 1BR3 cells showed 1.9 ± 0.5 micronuclei per 100 cells. This value was significantly higher in the HLEpiC cell line (5 ± 1; *p* < 0.01) while it was similar to that of the control in the HRPEpiC (1.9 ± 0.5; *p* > 0.5) (Figure 2). 

As detailed in the Introduction, the early γH2AX foci indicate the DSB recognized by NHEJ, while the residual γH2AX foci reflect the unrepaired DSB [26]. We applied γH2AX immunofluorescence to the untransformed human lens HLEpiC and retinal HRPEpiC. No significant yield of spontaneous γH2AX foci was observed in either HLEpiC or HRPEpiC cell lines. In the radioresistant skin fibroblasts, the number of γH2AX foci scored 10 min post-irradiation was 79 ± 6 per cell, consistent with the rate of 37 ± 4 per Gy per cell published previously [41,42]. The number of γH2AX foci scored 10 min post-irradiation in the two tested lens and retinal cell lines was found to be significantly lower than that in the radioresistant control (HLEpiC: 50 ± 0 and HRPEpiC: 65 ± 5 foci per cell; *p* < 0.05) (Figure 3A). As abundantly documented [21,26], these findings do not suggest that there were fewer DSB physically induced per nuclei in eye cells but rather that there were fewer DSB recognized by H2AX phosphorylation, suggesting an impairment in the DSB recognition step in agreement with the RIANS hypothesis [26]. It must also be stressed here that the size of nuclei cannot mathematically explain the differences between the γH2AX foci data. Between 10 min and 24 h post-irradiation, the number of γH2AX foci decreased in all the cell lines tested to reach a value that was non-significantly different from zero (Figure 3A). At 24 h post-irradiation, the human lens and the retinal epithelial cells showed a similar number of γH2AX foci as the radioresistant controls (1BR3: 1 ± 1; HLEpiC: 0.8 ± 0.6; and HRPEpiC: 1.5 ± 0.4 γH2AX foci per cell, *p* > 0.5). These last findings suggest that the repair rate of DSB recognized by NHEJ can be considered normal (Figure 3A). 

In order to consolidate the above hypothesis that less RI DSB are recognized by NHEJ after irradiation in lens cells, anti-*pATM* immunofluorescence was applied to the same cell lines with the same irradiation scenario. No spontaneous pATM foci were observed in the three cell lines tested. In the radioresistant skin fibroblasts, the number of pATM foci scored 10 min post-irradiation was 40 ± 4 per cell, in agreement with previous studies [28,29,30,31,32,33] (Figure 3B). Like with γH2AX foci, the number of pATM foci scored 10 min post-irradiation in the two tested lens and retinal cell lines was found to be significantly lower than that in the radioresistant control (HLEpiC: 23.7 ± 1.2 and HRPEpiC: 25 ± 5 foci per cell; *p* < 0.05), supporting a delay in the RIANS. The number of pATM foci decreased progressively to a value non-significantly different from zero at 24 h post-irradiation (Figure 3B). 

As briefly evoked in the Introduction, the MRE11 nuclease protein is a component of the RAD50–MRE11–NBS1 complex [42,43]. Interestingly, as described elsewhere, cells from cancer syndromes show either a maximal number of MRE11 foci from 10 min to 1 h after 2 Gy X-rays (e.g., retinoblastoma, tuberous sclerosis complex, and neurofibromatosis type 1) or the MRE11 foci are impaired (e.g., in *BRCA1*-, *BRCA2*-, *BLM*-, *FANC*-, and *ATM*-mutated cells) [33,43]. Conversely, the number of MRE11 foci progressively increases to reach its maximum 24 h post-irradiation in cells from aging syndromes (e.g., Hutchinson–Gilford progeroid syndrome, Huntington disease, and Usher syndrome) [33,43]. From the above findings, a delayed RIANS should impact the kinetics of the MRE11 foci. Hence, we assessed the number of MRE11 foci in HLEpiC and HRPEpiC cells with the same irradiation conditions as described above. No significant yield of spontaneous MRE11 foci was observed in the cell lines tested. In the radioresistant skin fibroblasts, the MRE11 foci varied from 2 to 8 h post-irradiation and reached their maximal yield at 4 h (7 ± 2 MRE11 foci per cell) (Figure 3C). The shape of the MRE11 foci kinetics in the human lens and retinal epithelial cells appeared to be clearly different. The HRPEpiC cell line showed a maximal number of 11.3 ± 4 MRE11 foci at 1 h post-irradiation and no foci at 24 h post-irradiation. The HLEpiC cell line showed a progressive increase in the number of MRE11 foci, with a maximum of 3.1 ± 2 MRE11 foci reached at 4 h post-irradiation. At 24 h post-irradiation, the number of MRE11 foci was different from zero and higher than in the control cells (1BR3: 2 ± 1, HLEpiCpiC: 5 ± 0; *p* < 0.05) (Figure 3C). 

### 3.2. Radiobiological Characterization of the Human Transformed HLE-B3 Lens Cell Line

We then examined the radiobiological features of the human-transformed HLE-B3 lens cell line with the same RIANS biomarkers as described above. The immortalized HLE-B3 cells showed a number of residual micronuclei 24 h post-irradiation that was significantly higher than that of HLEpiC (HLE-B3: 29 ± 0.5 vs HLEpiC: 5 ± 1 micronuclei per 100 cells, respectively; *p* < 0.0001) (Figure 4A). These results consolidate the gross genomic instability of the HLE-B3 cell line and justify our choice not to use it as a model for our investigations. The HLE-B3 cells showed significantly more γH2AX foci at 10 min after irradiation than the HLEpiC cell line (59 ± 1 vs 46.6 ± 3.3 γH2AX foci, respectively, *p* < 0.05). At 24 h post-irradiation, the HLE-B3 cell line showed a number of γH2AX foci significantly higher than the HLEpiC cell line (5.77 ± 0.7 vs 0.85 ± 0.6 foci, respectively, *p* < 0.0001) (Figure 4B), suggesting a DSB repair defect in the transformed HLE-B3 cell line. Furthermore, the HLE-B3 cell line showed a lower number of pATM foci 10 min after irradiation in comparison with the HLEpiC cell line (8.5 ± 1.5 vs 23.7 ± 1.2 pATM foci per cell, respectively, *p* < 0.01) (Figure 4C). However, care must be taken with regard to the pATM data since, in practice, the particular shape of pATM foci in the HLE-B3 cells may make their scoring difficult. Concerning the MRE11 foci, unlike the HLEpiC cell line, the HLE-B3 cell line showed a non-significant number of MRE11 foci from 10 min to 24 h after irradiation (0 ± 0 foci per cell at 10 min; 0.1 ± 0.1 foci per cell at 1 h; 0.5 ± 0.5 foci per cell at 4 h; and 1.1 ± 0.9 foci per cell at 24 h) (Figure 4D).

### 3.3. Attempts to Establish Human Epithelial Lens Cell Lines from Adults

Faced with the lack of availability of human lens epithelial cells, the low proliferating ability of the HLEpiC cell line, and the biases associated with the transformed HLE-B3 cell line, we attempted to establish human lens epithelial cell lines from adult eyes, and various procedures were considered:A first set of trials (Set 1) was carried out after adult cataract surgery. The trials consisted of obtaining lens epithelial cells from different conditions: One clear lens, one moderate cataractous lens, and one dense cataractous lens. Set 1 procedure was based on phacoemulsification, allowing the fragmentation of the lens and a UV probe to aspirate the cataract. This resulted in the destruction of some lens, and the challenge was to get a critical number of viable and clonogenic cells. However, this procedure failed to provide clones.A second set of trials (Set 2) consisted of obtaining whole lens from adult enucleation procedures. Set 2 procedure was based on the extraction of the whole lens, thus preserving the cells of interest. However, most of the donors were adults older than 50, and, again, this procedure failed to provide clones. Another critical step of the procedure was that only one layer of epithelial cells is present on the anterior side of the lens capsule, and breaking the capsule can cause the detachment of the cells from each other, therefore reducing the chances of cell proliferation.

### 3.4. Radiobiological Features of the Porcine Lens Cell Lines

The technical difficulty of establishing human lens epithelial cell lines prompted us to investigate the radiation response of lens cells in some animal models. There was an opportunity to work on the porcine lens, and we therefore established four cell lines to document their radiation response by applying the same experimental conditions and techniques as those described above and in Materials and Methods. In order to compare the radiobiological data from the lens, we used historical data obtained from a porcine skin fibroblast, PSF-1, established in the 1990s.

The number of micronuclei assessed 24 h after 2 Gy X-rays in the PSF-1 and in the four porcine lens (CP) cell lines was examined. The four porcine cell lines tested showed a significantly high number of micronuclei in comparison with the skin PSF-1 cell line (CP1: 12 ± 2.5; CP2: 15.8 ± 2.3; CP3: 14 ± 1.6 and CP8: 19.5 ± 2; PSF-1: 6 ± 3, respectively; *p* < 0.01). These results support significant RI genomic instability in the porcine lens epithelial cell lines tested (Figure 5).

Although PSF-1 fibroblasts showed lower yields of γH2AX foci from 10 min to 24 h post-irradiation, the γH2AX foci kinetics were not found to be different from the human skin 1BR3 fibroblast ones (*p* > 0.5). The number of γH2AX foci scored 10 min after 2 Gy in the radioresistant control PSF1 was 71 ± 3 per cell, while the corresponding numbers of γH2AX foci in the four porcine lens epithelial cells were significantly lower (CP1: 38 ± 0.09, CP2: 42 ± 1.5, CP3: 32 ± 3, and CP8: 42 ± 1.5 γH2AX foci per cell; *p* < 0.001) (Figure 6A), suggesting impaired DSB recognition. At 24 h post-irradiation, the four CP cell lines showed no significant difference in the number of γH2AX foci with the radioresistant control (*p* > 0.5), suggesting a normal repair rate of DSB recognized by NHEJ (Figure 6A). Altogether, these findings suggest a lack of DSB recognition by NHEJ but are not accompanied by an important DSB repair defect. 

With regard to the pATM foci, the number of pATM foci scored 10 min after 2 Gy X-rays in the radioresistant control PSF-1 was 34 ± 4 pATM foci (Figure 6B). The number of pATM foci assessed 10 min after 2 Gy X-rays in the four CP cell lines was significantly lower than that in the radioresistant fibroblast control (CP1: 21 ± 0.06, CP2: 23 ± 1.4, CP3: 20 ± 1.5, and 20 ± 1.7 pATM foci per cell, *p* < 0.05) (Figure 6B). These data suggest that CP cells showed abnormal ATM nuclear kinase activity in response to radiation (Figure 6B).

With regard to the MRE11 foci, the PSF-1 cells showed a maximum of 5 MRE11 foci per cell 1 h post-irradiation and 3 MRE11 foci per cell 4 h post-irradiation. Three of the four CP cell lines tested showed a curvilinear increase in the number of MRE11 foci per cell at 4 h post-irradiation (CP1: 1, CP2: 2, and CP8: 2 MRE11 foci per cell) (Figure 6C). These findings indicate that the porcine lens epithelial cells, like the human lens epithelial cell line HLEpiC tested, show an MRE11 kinetics similar to those of cells from aging syndromes. The CP3 cell lines did not show any significant MRE11 foci (Figure 6C).

### 3.5. Identification of X-Proteins Specific of Lens Cells 

Our findings showed that the lens cells tested systematically elicit delayed RIANS with fewer γH2AX and pATM foci than expected. As evoked in the Introduction, the delay of RIANS is generally due to proteins overexpressed in the cytoplasm that are substrates of ATM phosphorylation (i.e., holding putative SQ or TQ domains) [26]. We, therefore, made the hypothesis that X-proteins specific to the lens may facilitate the aging process and cataractogenesis. In order to identify X-proteins specific to the lens, we performed immunofluorescence and immunoblots to select some X-protein candidates. As a first step, we investigated various crystallin proteins since they are the major lens components [44]. However, the three investigated proteins, β-crystallin B1, αA-crystallin, and αB-crystallin, have no or few putative SQ/TQ sites on their sequences (1 SQ/0 TQ, 0 SQ/0 TQ, and 0 SQ/0 TQ, respectively). No PLA dots (representing an interaction between ATM and these proteins) were detected in the human lens epithelial HLEpiC cells. We then focused on two highly expressed proteins in the lens with an important number of putative SQ/TQ sites and whose mutation causes the development of cataracts: Pax6 (3 SQ/2 TQ) and FYCO1 (10 SQ/6 TQ). However, these two proteins appeared to be nuclear, whatever the conditions. 

As a second step, we focused on the BFSP2 protein (3 SQ and 2 TQ). We first examined BFSP2 expression by immunofluorescence. BFSP2 appeared to be mainly cytoplasmic, with some nuclear forms. Interestingly, we observed some perinuclear crowns of BFSP2 staining before irradiation, disappearing up to 1 h after irradiation, and then reappearing 4 h after irradiation (Figure 7).

We thereafter investigated the BFSP2 expression in cytoplasmic protein extracts of the fibroblast radioresistant control cell line (MRC5) and of the four porcine lens epithelial cell lines (CP1, CP1, CP3, and CP8) by applying immunoblots (Figure 8). Cell extracts were performed prior to irradiation and 10 min, 1 h, 4 h, and 24 h after 2 Gy X-ray exposure. Immunoblots showed abundant cytoplasmic forms of the BFSP2 protein in all cell lines tested (Figure 8). However, this expression varied in some cell lines as a function of post-irradiation time. 

Since the RIANS is delayed in the porcine lens epithelial cell lines and since the BFSP2 protein was found to be very abundant in the cytoplasm, we hypothesized that some BFSP2–ATM complexes may exist. The PLA assay was applied to the CP cell lines (Figure 9). The PLA assay revealed significantly more cytoplasmic ATM–BFSP2 in the CP cell lines than in the human skin fibroblast control (*p* < 0.001). The influence of the irradiation on the number of PLA foci reflecting the ATM–BFSP2 complexes was cell line dependent. 

### 3.6. Perinuclear pATM Crowns in Lens Epithelial Cells 

Recently, in this journal, we have pointed out that fibroblasts from patients suffering from Alzheimer’s disease (AD) specifically show spontaneous perinuclear pATM crowns [34]. The perinuclear pATM crowns were found to be the result of the progressive accumulation of ATM monomers in response to permanent endogenous and exogenous oxidative stress. Such accumulation was facilitated by overexpression of the perinuclear forms of APOE protein and the formation of the ATM–APOE complex that composes the inner layer of the pATM crowns. Above this first layer, the ATM monomers accumulate by forming ATM dimers [34]. As evoked above, most of the proteins whose mutations cause neurodegenerative diseases and aging syndromes are situated inside the nuclear membrane or around the nucleus [34]. Since these proteins, such as APOE for AD, are also potential ATM phosphorylation substrates, they may serve as X-proteins. Hence, the ATM–X protein complexes may be perinuclear and favor the agglutination of ATM monomers around the nucleus, which facilitates the formation of perinuclear pATM crowns. Since the number of perinuclear pATM crowns was found to increase with cell culture passages that mimic aging in cells [34], we therefore examined the formation of perinuclear pATM crowns in CP cells. Interestingly, perinuclear pATM crowns were found to be thicker and more numerous in the highest cell culture passages and reached their maximal value at passage 12 on average (Figure 10).

The percentage of cells with perinuclear pATM crowns was plotted against the number of cell passages, and some individual differences appeared. Mathematically, such differences cannot be due to errors in cell seeding or passage number calculation. The highest regression coefficients were obtained when the data were fitted to exponential and power laws (Figure 11A). By considering exponential law, the percentage of perinuclear pATM crowns was found to not be at passage 0, suggesting an unequal predisposition to aging between cells, ranging from 1.8 to 9.1 perinuclear pATM crowns per 100 cells. The cell doubling period T, calculated from a = ln2/T, ranged from 2 to 8 passages. The fact that the maximal value of perinuclear pATM crowns was reached at passage 12 corresponds to a cell progression from 1 to 2^12^ = 4096 cells if one passage corresponds to one doubling time and from 1 to 2^24^ = 16,777,216 cells if one passage corresponds to two doubling times. Altogether, these findings are consistent with our model of accelerating aging with the agglutination of ATM around the nucleus. Further investigations are, however, needed to relate these findings to the formation of pATM crowns with the BFSP2 protein that would be phosphorylated by ATM in an inner layer of the crown, as recently observed in AD cells with the APOE protein [34].

In the context of the RIANS-aging model, exposure to ionizing radiation may lead to a transient monomerization of ATM and therefore to a temporary disappearance of the perinuclear pATM crown [34]. In order to investigate the relationship between the number of perinuclear pATM crowns, the dose, and repair time, we scored the pATM crowns after 0.1 to 10 Gy at post-irradiation times ranging from 10 min to 72 h. By considering the data assessed 10 min and 24 h post-irradiation, we obtained sigmoidal curves for both CP1 and CP8 data, suggesting that the pATM crowns do not significantly disappear in the dose range of 0.1–0.2 Gy, decrease progressively from 0.5 to 1 Gy, and form a plateau of minimal values from 1 to 10 Gy. This trend was found to be quite similar with both CP cell lines tested (Figure 11B). The dose values corresponding to a 50% decrease ranged from 0.76 to 0.88 Gy in both cell lines, whatever the post-irradiation time (Figure 11B).

## 4. Discussion

### 4.1. Too Few Cell Lines but a Common Conclusion

This work aims to document the radiation response of human epithelial lens cells. However, the radiobiological characterization of this type of cell remains limited by the lack of availability of human epithelial lens cell lines, notably explained by the difficulties in amplifying them to high cell culture passages. By applying the RIANS biomarkers (γH2AX, pATM, and MRE11) to the HLEpiC cell line from ScienCell (Carlsbad, CA, USA), a significant delay appeared in RIANS, suggesting that this lens cell line elicits moderate but significant radiosensitivity. Furthermore, the MRE11 foci kinetics found in HELpiC cells showed an analogy with those obtained from skin cells from aging syndromes, with a maximal value reached at 24 h post-irradiation. The radiobiological features of the transformed B3 lens cell line led to the same conclusions. Hence, while we are aware that too few human lens cell lines have been tested, our findings and literature data converge on the fact that (1) the human lens is a radiosensitive tissue [45] and (2) the culture and establishment of a primary cell line are limited by a small number of passages before reaching senescence. However, care must be taken about (1) the effect of immortalization on the molecular and cellular radiation responses [46]. In addition, Hamada reported the impossibility of transforming the HELpiC cell line with hTert; (2) as mentioned above, the ScienCell Research Laboratories (Carlsbad, CA) proposed several lots of primary normal human lens epithelial cells derived from different donors (likely from fetal origin) under the same name (HLEpiC). Hamada has reported the first estimation of the cellular radiosensitivity of HLEpiC (renamed HLEC1) associated with a permanent cell cycle arrest [39]. The consecutive failures to establish a human epithelial lens cell line from adult materials prompted us to use porcine lenses.

In addition, we identified a potential X-protein specific to the lens, i.e., an overexpressed cytoplasmic ATM substrate whose perinuclear sub-localization may facilitate the progressive agglutination of ATM around the nucleus and the formation of the perinuclear pATM crown. Lastly, in porcine lens cells, the number of perinuclear pATM crowns reached its maximal value at cell culture passage 12, which supports a limited lifespan in cell culture for this type of cell. The identification of a lens-specific X-protein (BFSP2), that is perinuclear, supports once again that the lens is a radiosensitive and radiodegenerative tissue.

### 4.2. Radiosensitivity of Lens: A Question of Radiobiology and of Semantics

The lens was considered one of the most radiosensitive tissues, such as the breast, thyroid, or skin [45,47]. However, such a statement reflects well the semantic confusion around the term “radiosensitivity”. Indeed, a systematic word analysis of the expressions, including the term “radiosensitivity”, in the ICRP reports revealed a long history of confusion likely due to the lack of a univocal definition of “radiosensitivity” [6]. For example, when eyes are said to be “radiosensitive”, it relates to RI cataracts, i.e., as a consequence of an RI cellular aging, while when breasts are said to be radiosensitive, it refers to RI breast cancer, i.e., as a consequence of an RI cellular transformation. Similarly, when skin is said to be radiosensitive, it is to describe radio-dermatitis, a consequence of RI toxicity and cellular death [6,7]. Hence, while the same term “radiosensitivity” is used to describe them, it must be stressed that RI cellular aging, transformation, and death processes do not involve the same pathways, do not obey the same dose responses, and do not require the same endpoints and biomarkers. Furthermore, these notions may be independent of each other [33,43]. For example, some genetic syndromes may be associated with RI adverse tissue reactions and RI cancers (e.g., Li–Fraumeni syndrome) or with RI adverse tissue reactions and RI aging (e.g., Werner’s syndrome). For all these reasons, we have proposed in 2016 the following definitions [7]: -“radiosensitivity”: any RI clinical and cellular events attributable to cell death (e.g., tissue reactions);-“radiosusceptibility”: RI cancers or any RI event attributable to cell transformation;-“radiodegeneration”: any aspect of the response to ionizing radiation attributable to aging.

As a consequence, the lens may be considered a radiosensitive and radiodegenerative but not radiosusceptible tissue (there is no report of cancer of the lens). 

### 4.3. Lens and the RIANS Model

Like with a number of genetic syndromes in our previous reports [28,29,30,31,32,33,48,49], the radiobiological characterization of lens cell lines was performed with γH2AX, pATM, and MRE11 immunofluorescence to build a unified and coherent mechanistic model [21,25,33]: As evoked in the Introduction, the RIANS model has provided a unified biological interpretation of numerous radiobiological phenomena such as hormesis [50], adaptive response [51], and hypersensitivity to low dose [25]. In the framework of the RIANS model, each level of oxidative stress corresponds to a specific number of induced DSB and a specific number of cytoplasmic monomerized ATM. For each genetic disease, some proteins specifically overexpressed in the cytoplasm and interacting with the ATM protein may sequester the ATM monomers in the cytoplasm and prevent their diffusion in the nucleus and the phosphorylation of H2AX histones at the DSB sites. These proteins have been called X-proteins. They reflect the individual predisposition to an abnormal response to ionizing radiation [33,43].

Interestingly, by applying the same approach to the lens, we have identified at least one X-protein for the lens, the BSP2 protein, that shows perinuclear sublocalization and therefore may favor the formation of perinuclear pATM crowns in response to permanent stress. In the framework of the RIANS model, such proteins may be considered tissue-specific X-proteins, unlike the “syndromic” X-proteins defined in skin fibroblasts and specific to each human genetic disease. Hence, the lens appears to be a tissue whose radiosensitivity belongs to group II defined by the RIANS model (delayed RIANS, moderate radiosensitivity, predisposition to cancer or else to aging) [33,43] (Figure 12).

By combining the radiobiological characterizations accumulated with skin fibroblasts from genetic diseases associated with juvenile cataracts and those of the lens cells, the syndromic and lens-specific tissue X-proteins may contribute together to the sequestration of the ATM monomers in lens cells to explain both individual and tissue predispositions to aging. In other terms, the syndromic X-proteins may accelerate a process already “programmed” in the lens thanks to tissue X-proteins such as BFSP2 (Figure 12). Our findings are the first example of both individual and tissue predispositions to abnormal responses to IR interpreted by the RIANS model (Figure 12).

### 4.4. Quantitative Features Related to RI aging of Lens Cells

By considering the perinuclear pATM crowns as specific biomarkers of aging [34], Figure 11A clearly shows that the number of perinuclear pATM crowns per 100 cells increases as a function of cell culture passages at different rates according to the CP cell line. All the findings converge to a maximal value at around passage 12. However, it is noteworthy that, after this passage, some preliminary data suggest that a horizontal plateau is reached for three additional passages before the number of cells with pATM crowns decreases drastically. While further investigations are needed to document the variation of the number of cells with perinuclear crowns after passage 12, it must be reminded that cells with pATM crowns can accumulate with time a certain amount of unrepaired DNA strand breaks that should lead to senescence death. Hence, a decrease in the average number of cells with pATM crowns is therefore not surprising after a certain number of passages, inasmuch as in vitro cultured cells can be considered synchronized. Again, experiments are in progress to validate this hypothesis.

Exposure to ionizing radiation not only adds more DNA breaks in the nucleus but also triggers the monomerization of the ATM proteins that help recognize them. It has been suggested that, while about 40 DSB are induced per Gy per fibroblast, about 10^5^ ATM monomers per Gy are produced in the cytoplasm [25]. Since the pATM crowns are composed of ATM dimers recognized by the anti-*pATM* antibodies, exposure to ionizing radiation may lead to the monomerization of the ATM dimers that compose the pATM crowns and therefore may cause them to disappear. However, exposure to ionizing radiation may not cause a definitive but a transient steric dispersion of the ATM monomers around the nucleus, and their local concentration may facilitate again the formation of ATM dimers after irradiation and therefore that of the pATM crowns. Hence, after irradiation and a “relaxation time” that would depend on the dose, the pATM crowns may reappear. In our conditions, the relaxation time deduced from our preliminary data was found to be between 24 h and 48 h. 

In order to relate our findings to RI cataractogenesis, further experiments are needed to link the number of cells with pATM crowns, the number of cells undergoing senescence, and the in vivo occurrence of cataracts. However, an interesting notion emerges from our findings: The predisposition to accelerated aging and to early cataracts that would be reflected by a non-negligible number of cells with pATM crowns at the lowest passages. Interestingly, Figure 11B suggests that the equivalent of 2 Gy followed by 10 min post-irradiation leads to a complete disappearance of the pATM crowns. Such findings may indicate that pATM crowns may be the result of the cumulative agglutination of the corresponding amount of ATM monomers around the nucleus, i.e., about 2 × 10^5^ ATM monomers. For the formation of the first monolayer, let us consider an average nucleus radius of lens cells of 5–10 μm, an average surface of the nuclear membrane of about 800 μm^2^, and an average number of nuclear pores of about 1000. The formation of a pATM crown monolayer may therefore require at least the agglutination of about 200 ATM monomers per nuclear pore and 250 ATM monomers per μm^2^. 

Obviously, different scenarios of irradiation may fill such requirements (high-dose flash exposure, chronic and repeated doses, irradiation at a low dose rate, permanent oxidative stress) and are the subject of calculations in progress. 

## 5. Conclusions

Altogether, our findings support that all the epithelial lens cells tested, whether human or porcine, show a delay in the radiation-induced ATM nucleoshuttling, late MRE11 foci, and perinuclear ATM crowns, as already observed in cells from aging syndromes [34]. In the lens, the formation of the perinuclear ATM crowns is consistent with the overexpression of the BFSP2 protein, which localizes around the nucleus and holds the SQ/TQ putative domains of phosphorylation by ATM. Further investigations are, however, needed to relate the formation of cataracts in vivo, their nature, their occurrence rate, their specificities with regard to the radiation type, and the values of the above molecular and cellular endpoints depending on the ATM kinase activity involved in the RIANS model.

## Figures and Tables

**Figure 1 cells-12-02118-f001:**
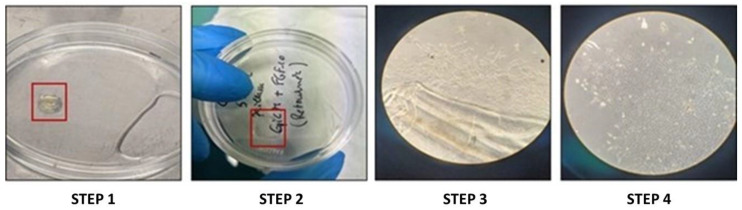
Establishment of porcine lens epithelial cell lines. (STEP 1) consisted of placing the lens (red square) in the Petri dish with the anterior side down. The capsule was then opened after an incision was made on its posterior side. (STEP 2). The fibrous mass was discarded, and the capsule was incubated with the cells facing up. A few days after the incubation, cells started to emerge (STEP 3) from the capsule, and a few days later colonies were formed and cell lines were established. (STEP 4).

**Figure 2 cells-12-02118-f002:**
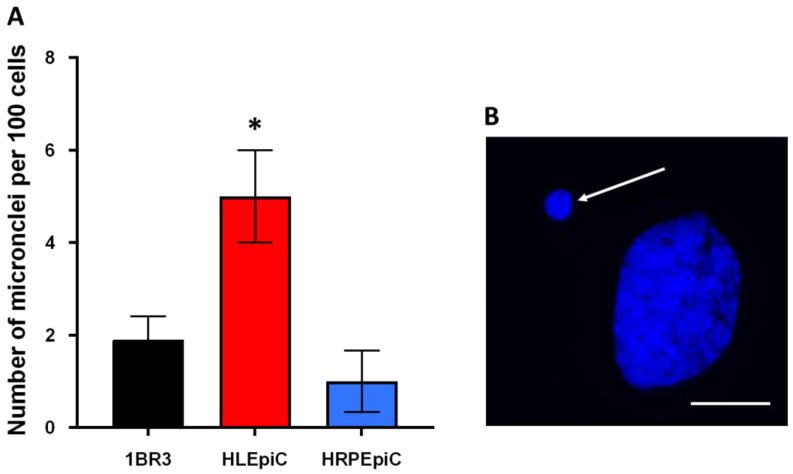
Micronuclei in human lens and retinal epithelial cell lines. (**A**) Number of micronuclei per 100 cells assessed 24 h post-irradiation in the radioresistant control 1BR3, the human lens epithelial cell line HLEpiC, and the human retinal pigment epithelial cell line HRPEpiC. Each plot represents the mean ± standard error of the mean (SEM) of three replicates. (**B**) The image shows representative example of a micronucleus (white arrow) observed with DAPI counterstaining. The white bar corresponds to 10 μm. The asterisk corresponds to a *p* < 0.05 difference by comparison with 1BR3 data.

**Figure 3 cells-12-02118-f003:**
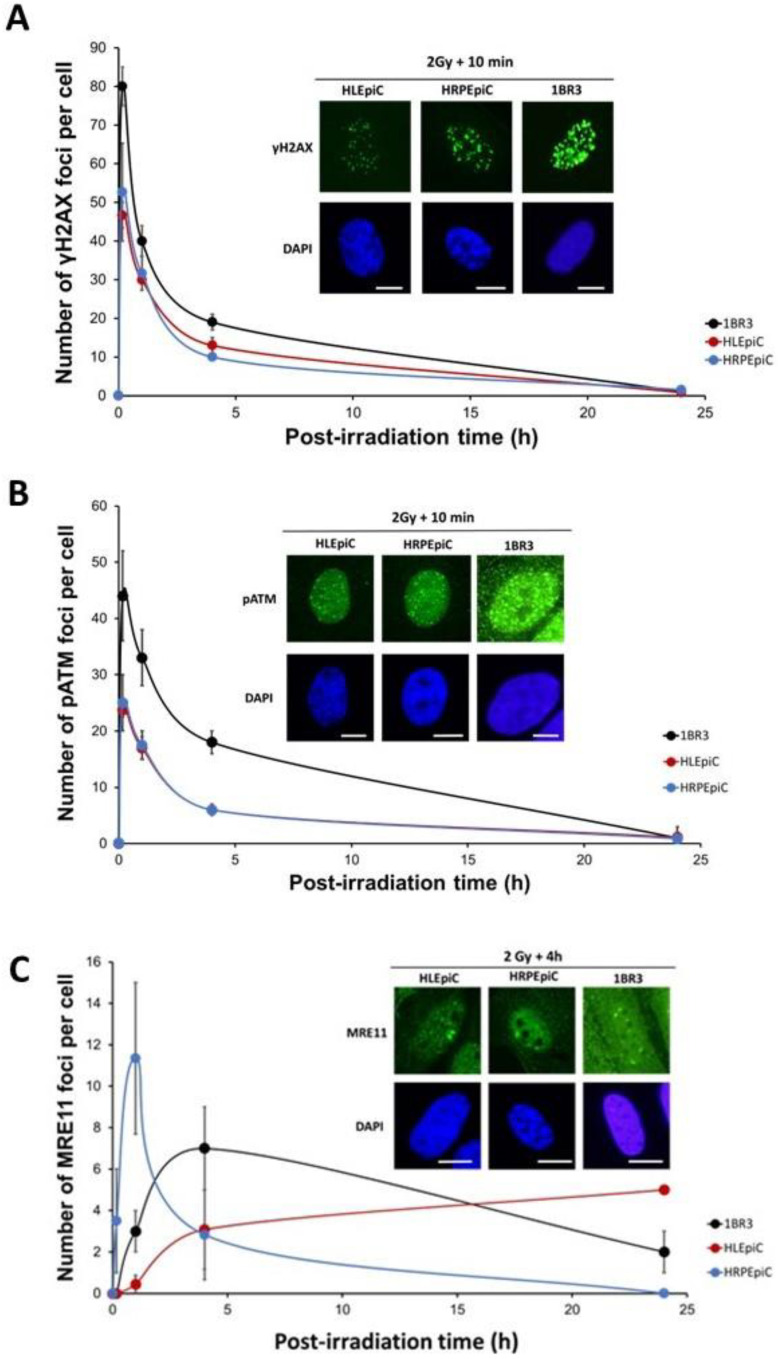
Kinetics of γH2AX, pATM, and MRE11 foci in untransformed human lens and retinal epithelial cells. (**A**) Anti-γH2AX immunofluorescence was applied to the radioresistant fibroblast control 1BR3, the lens epithelial cell line HLEpiC, and the retinal pigment epithelial cell line HRPEpiC. The number of γH2AX foci was plotted against the post-irradiation time. Each plot represents the mean ± SEM of three replicates. The insert shows a representative image of DAPI-stained nuclei and γH2AX foci observed at 10 min post-irradiation (2 Gy X-rays) in each indicated cell line. The white bar corresponds to 10 μm. (**B**) Anti-*pATM* immunofluorescence was applied to the radioresistant fibroblast control 1BR3, the lens epithelial cell line HLEpiC and the retinal epithelial cell line HRPEpiC. The number of pATM foci was plotted against the post-irradiation time. Each plot represents the mean ± SEM of three replicates. The insert shows a representative image of DAPI-stained nuclei and pATM foci observed at 10 min post-irradiation (2 Gy X-rays) in each indicated cell line. The white bar corresponds to 10 μm. (**C**) Anti-*MRE11* immunofluorescence was applied to the radioresistant fibroblast control 1BR3, the lens epithelial cell line HLEpiC and the retinal epithelial cell line HRPEpiC. The number of MRE11 foci was plotted against post-irradiation time. Data were obtained from the radiosensitive fibroblast control 1BR3, the lens epithelial cell line HLEpiC and the retinal pigment epithelial cell line HRPEpiC. Each plot represents the mean ± SEM of three replicates. The insert shows a representative illustration of DAPI-stained nuclei and MRE11 foci observed at 4 h post-irradiation in each indicated cell line. The white bar corresponds to 10 μm.

**Figure 4 cells-12-02118-f004:**
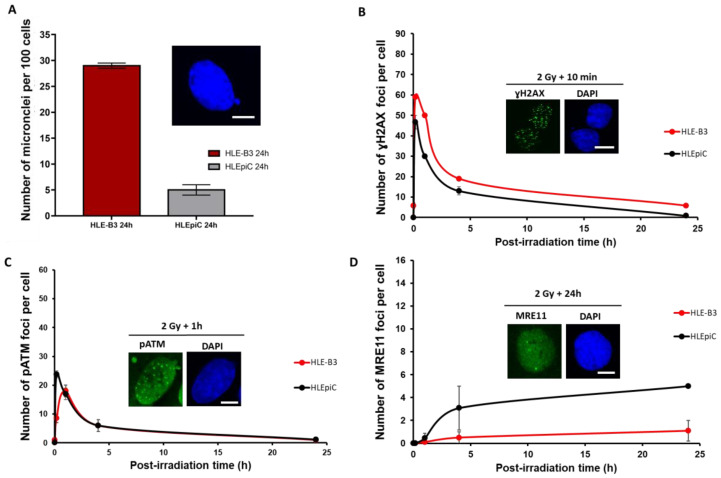
Comparison of the radiobiological features of the HLE-B3 and HLEpiC cell lines. (**A**) The number of micronuclei per 100 cells assessed 24 h after 2 Gy X-rays in the indicated cell lines. Each plot represents the mean ± SEM of three replicates. The insert shows a representative illustration of DAPI-stained micronucleus in HLE-B3 cells. The white bar corresponds to 10 μm. (**B**) Anti-*γH2AX* immunofluorescence was applied to the indicated cell lines. The number of γH2AX foci was plotted against the post-irradiation time. Each plot represents the mean ± SEM of three replicates. The insert shows a representative image of DAPI-stained nuclei and γH2AX foci observed at 10 min post-irradiation in HLE-B3 cells. The white bar corresponds to 10 μm. (**C**) Anti-*pATM* immunofluorescence was applied to the indicated cell lines. The number of pATM foci was plotted against the post-irradiation time. Each plot represents the mean ± SEM of three replicates. The insert shows a representative image of DAPI-stained nuclei and pATM foci observed at 1 h post-irradiation in HLE-B3 cells. The white bar corresponds to 10 μm. (**D**) Anti-*MRE11* immunofluorescence was applied to the indicated cell lines. The number of MRE11 foci was plotted against post-irradiation time. Each plot represents the mean ± SEM of three replicates. The insert shows a representative image of MRE11 observed at 24 h in HLE-B3 cells. The white bar corresponds to 10 μm.

**Figure 5 cells-12-02118-f005:**
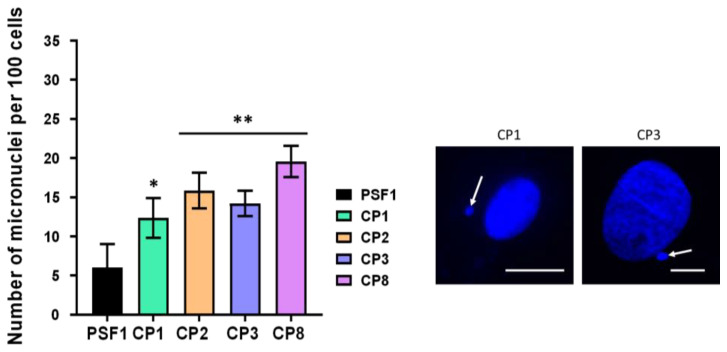
Number of residual micronuclei in porcine lens epithelial cell lines. The number of micronuclei per 100 cells was assessed 24 h after 2 Gy X-rays in the PSF-1 and the CP1, CP2, CP3, and CP8 cell lines. Each plot represents the mean ± SEM of five replicates. The insert shows representative example of micronuclei (white arrow) observed with DAPI counterstaining in CP1 and CP3 cells. The white bar corresponds to 10 μm. One and two asterisks correspond to a *p* < 0.05 and *p* < 0.01 difference by comparison with PSF1 data, respectively.

**Figure 6 cells-12-02118-f006:**
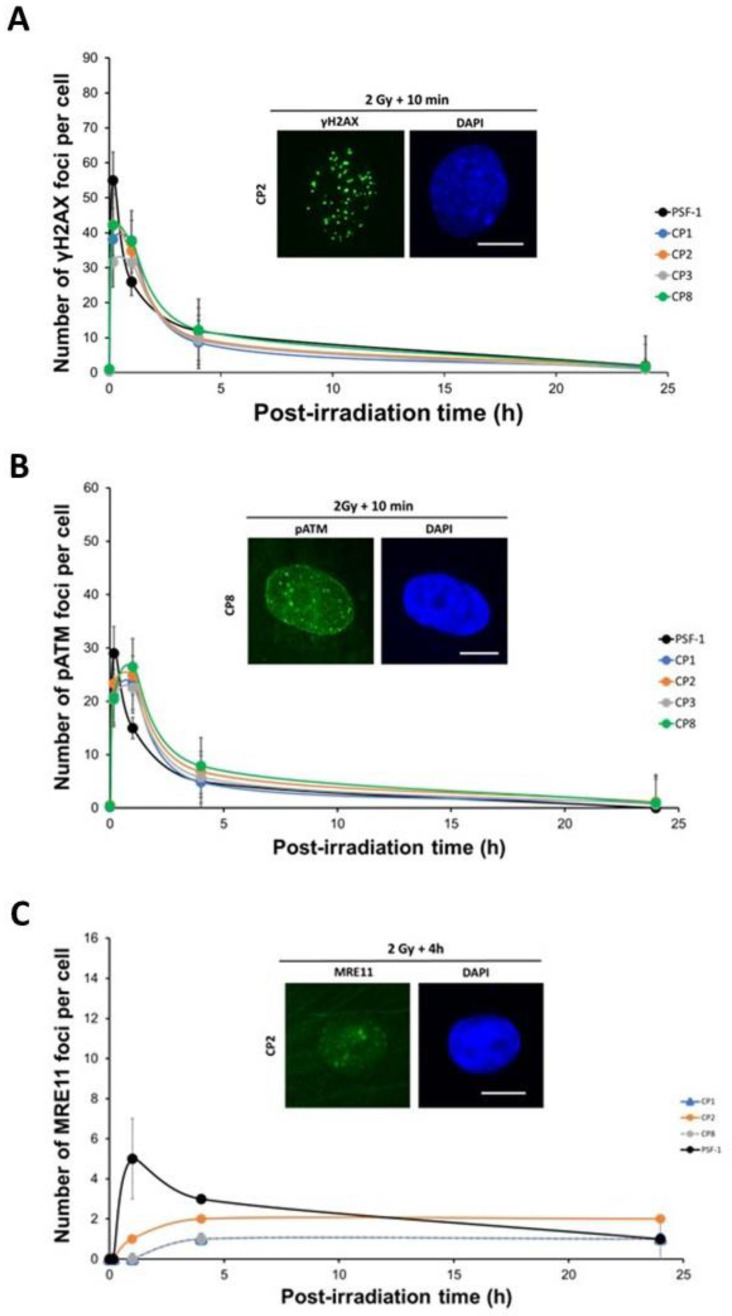
Kinetics of γH2AX, pATM, and MRE11 foci in porcine lens epithelial cells. (**A**) Anti-*γH2AX* immunofluorescence was applied to the PSF-1 and the CP cell lines. The number of γH2AX foci was plotted against the post-irradiation time. Each plot represents the mean ± SEM of five replicates. The insert shows a representative image of DAPI-stained nuclei and γH2AX foci observed at 10 min post-irradiation (2 Gy X-rays) in CP2 cells. The white bar corresponds to 10 μm. (**B**) Anti-*pATM* immunofluorescence was applied to the PSF-1 and the CP cell lines. The number of pATM foci was plotted against the post-irradiation time. Each plot represents the mean ± SEM of five replicates. The insert shows a representative image of DAPI-stained nuclei and pATM foci observed at 10 min post-irradiation (2 Gy X-rays) in CP8 cells. The white bar corresponds to 10 μm. (**C**) Anti-*MRE11* immunofluorescence was applied to the PSF-1 and the CP cell lines. The number of MRE11 foci was plotted against post-irradiation time. Each plot represents the mean ± SEM of five replicates. The insert shows a representative image of DAPI-stained nuclei and MRE11 foci observed at 4 h post-irradiation in CP2 cells. The white bar corresponds to 10 μm.

**Figure 7 cells-12-02118-f007:**
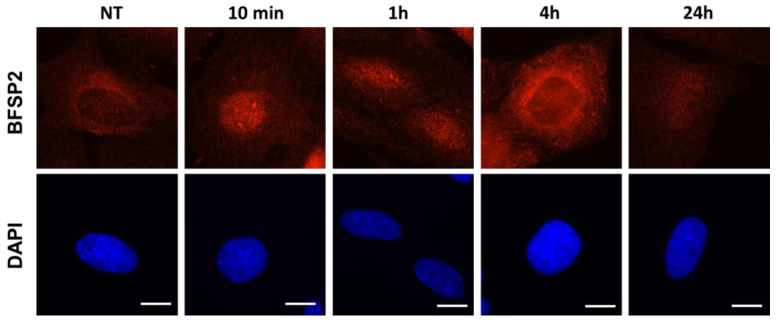
Subcellular localization of BFSP2 in porcine lens epithelial cells. Anti-*BFSP2* immunofluorescence was applied to the four CP cell lines either before or at the indicated post-irradiation times; only the CP8 cell line is represented in this figure since the other cell lines present similar data. The white bars represent 10 μm.

**Figure 8 cells-12-02118-f008:**
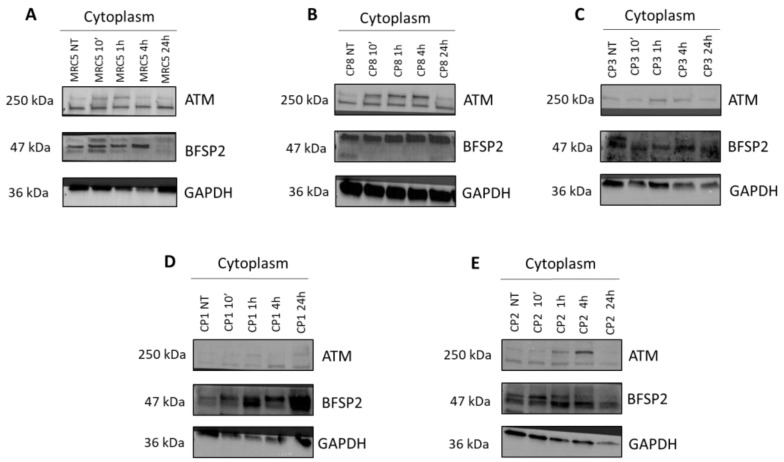
Representative immunoblots of BFSP2 protein expression in porcine lens epithelial cell lines. (**A**–**E**) Anti-*BFSP2* immunoblots with cytoplasmic protein extracts of the indicated control fibroblasts (MRC5) and porcine lens epithelial cell lines (CP1, CP2, CP3, and CP8), before and 10 min, 1 h, 4 h, and 24 h after 2 Gy X-rays.

**Figure 9 cells-12-02118-f009:**
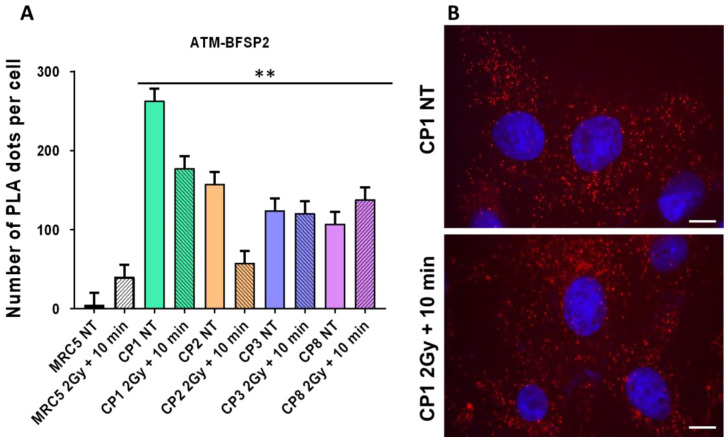
Interaction between ATM and BFSP2. (**A**) A proximity ligation assay (PLA) was applied to the indicated cell lines. The average number of red dots representing the cytoplasmic ATM–BFSP2 protein complexes per 100 cells was scored before (non-treated, NT) and 10 min after 2 Gy irradiation. The asterisks indicate a *p* < 0.01 difference by comparison with MRC5 NT data. Each plot represents the mean ± SEM of three replicates. (**B**) Representative PLA images obtained from the CP1 cell line. The nuclei were counterstained with DAPI (blue). The white bars represent 10 μm. Two asterisks correspond to *p* < 0.01.

**Figure 10 cells-12-02118-f010:**
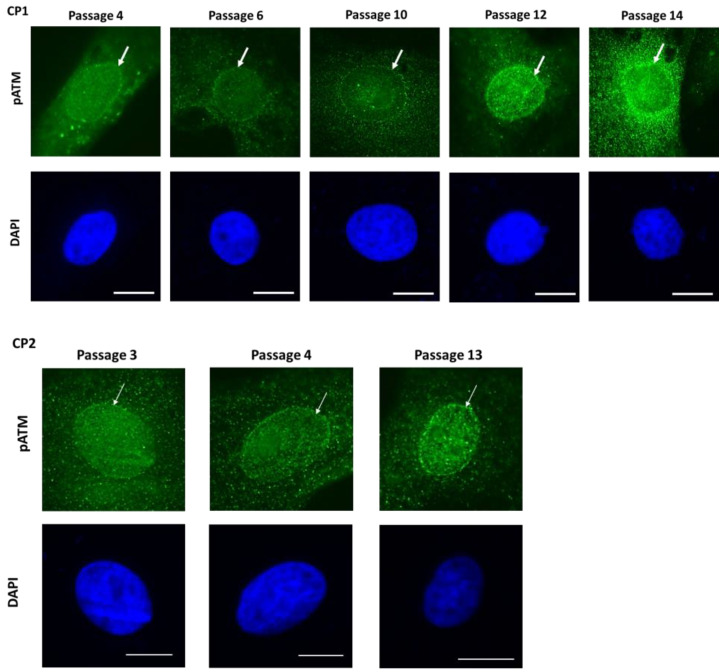
Spontaneous perinuclear localization of pATM in porcine lens epithelial cells. Representative immunofluorescence images of the perinuclear pATM crowns (white arrow) in the porcine lens epithelial cell lines CP1 and CP2 showing the progression of the crown size as a function of the cell culture passage. DAPI signals served as nuclear counterstaining. The white bars represent 10 μm.

**Figure 11 cells-12-02118-f011:**
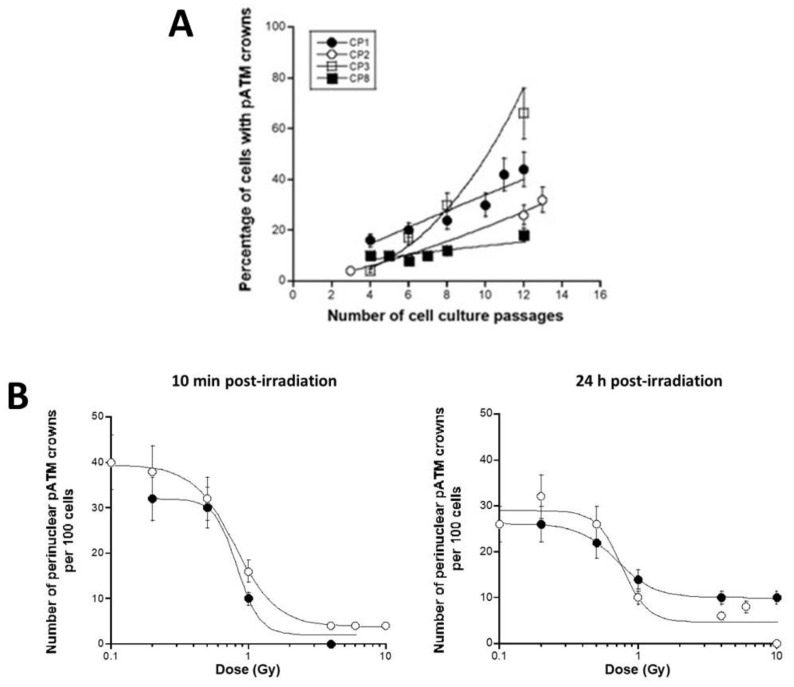
Specific features of the perinuclear pATM crowns observed in CP cells. (**A**) Percentage of cells with perinuclear pATM crowns as a function of cell culture passages for the indicated cell lines. Each plot corresponds to the mean ± SEM of two replicates. Data fit to an exponential law provides the following formulas for CP1, CP2, CP3, and CP8: y = 0.15 exp(0.129x), r = 0.977; y = 2.14 exp(0.207x), r = 1; y = 1.8 exp(3.18x), r = 0.97; y = 6.1 exp(0.083x), r = 0.92; respectively. (**B**) Percentage of cells with perinuclear pATM crowns as a function of dose and time. The percentage of cells with perinuclear pATM crowns was assessed after the indicated dose, followed by 10 min (left panel) or 24 h (right panel). Each plot corresponds to the mean ± SEM of two replicates for CP1 (white circles) and CP8 (closed circles). Data were fitted to a sigmoidal law.

**Figure 12 cells-12-02118-f012:**
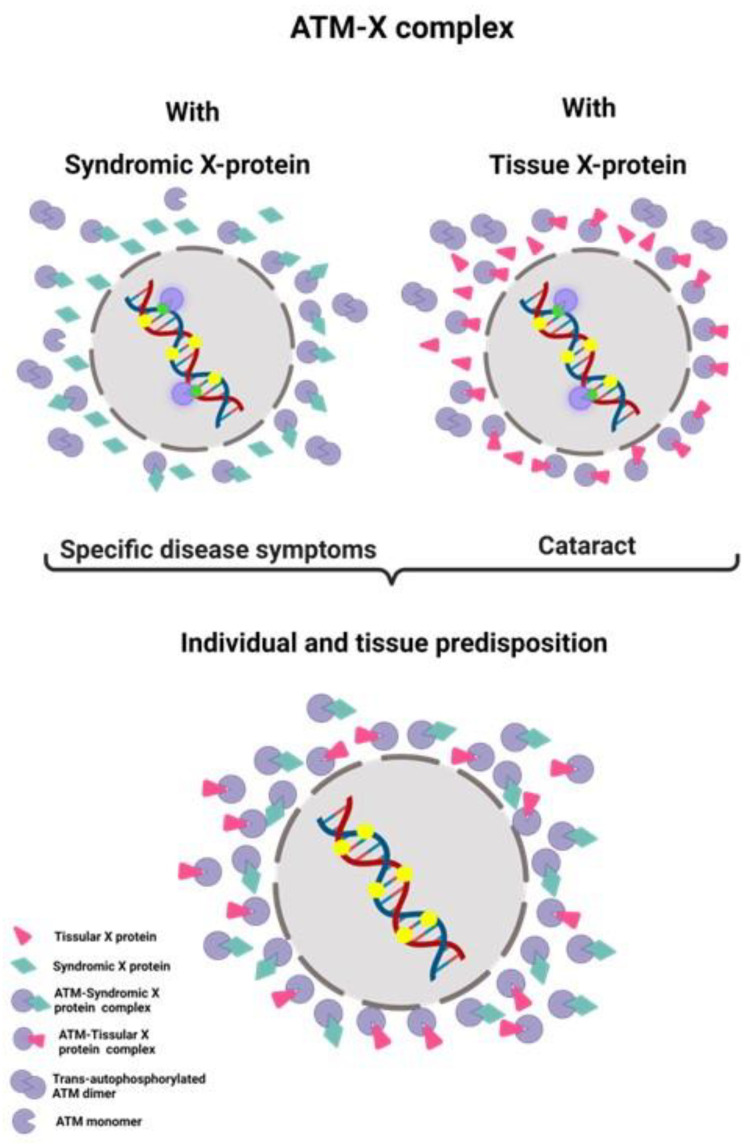
Mechanistic model for the mode of action of “syndromic” and “tissue” X-proteins. Once ATM dimers are monomerized in response to ionizing radiation, ATM monomers diffuse into the nucleus and trigger DSB recognition via H2AX phosphorylation. In some syndromes, the mutated protein responsible for the disease may also be an ATM substrate (syndromic X-protein) and thus prevent ATM monomers from diffusing into the nucleus by forming complexes with ATM. In other cases, one or several proteins overexpressed specifically in the tissue considered (the lens in our case) as tissue X-protein, may also be an ATM substrate and prevent its diffusion into the nucleus by forming complexes with ATM. Both syndromic and tissue X-proteins may contribute together to accelerated aging and the formation of perinuclear pATM crowns (Created with BioRender.com).

**Table 1 cells-12-02118-t001:** Average nucleus surfaces of the different cell lines used in this study.

Cell Line	HLEpiC	HRPEpiC	HLE-B3	1BR3	PSF-1	CP1	CP2	CP3	CP8
Nucleus surface (μm^2^)	1304 ± 82	964 ± 48	1156 ± 116	840 ± 52	780 ± 50	852 ± 104	848 ± 96	1044 ± 84	1008 ± 88

## Data Availability

All the data can be provided upon reasonable request.

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
