# Peer review of "The Radiobiological Characterization of Human and Porcine Lens Cells Suggests the Importance of the ATM Kinase in Radiation-Induced Cataractogenesis"

_cells, 2023, doi:10.3390/cells12162118_

Round 1

Reviewer 1 Report

This study has employed cultured human and porcine lens epithelial cells to investigate X-irradiation induced ataxia telangiectasia mutated nucleoshuttling (RIANS) in the cells. The data suggest a role for ataxia telangiectasia mutated (ATM) protein in X-irradiation induced cataractogenesis.

Following WWII, X-ray cataract was studied for over 50 years using a variety of animal models (reviewed by S. Lerman, Cataracts: Chemistry, Mechanisms and Therapy, 1964, pp. 137-141). The work showed how a compromised lens epithelium can lead initially to posterior subcapsular cataract, and ultimately to complete lens opacity. The work showed that X-ray affected only proliferating epithelial cells present in the equatorial region of the lens. Using shielding, it was shown that if the germinative region of the lens was protected, very high doses of X-ray could be passed through the lens center without causing any loss of transparency since cells in the lens center do not divide (Pirie A and Flanders PH, AMA Arch Ophthalmol., 1957). Also, young lenses had a much greater susceptibility to X-ray cataract compared to older lenses because of their more rapidly dividing population of epithelial cells (Cogan DG and Donaldson DD, AMA Arch. Ophthalmol., 1951).  One wonders how all of this might have affected the results of the current study, based on what stage of the cell cycle the six different cell lines were in when irradiated.

This manuscript was difficult for this reviewer to comprehend. It appeared that a considerable amount of false starts and negative data were included in the paper that could have been left out in order to focus more on the main finding of the study. Only a small proportion of the data contained in the 17 figures and one table of the paper are mentioned in the abstract.  

The incorrect use and non-use of the article "the" is frequent in the manuscript.

Author Response

Replies to the reviewer 1 ‘s comment

We thank the reviewer’s for his/her comments

This study has employed cultured human and porcine lens epithelial cells to investigate X-irradiation induced ataxia telangiectasia mutated nucleoshuttling (RIANS) in the cells. The data suggest a role for ataxia telangiectasia mutated (ATM) protein in X-irradiation induced cataractogenesis.

Following WWII, X-ray cataract was studied for over 50 years using a variety of animal models (reviewed by S. Lerman, Cataracts: Chemistry, Mechanisms and Therapy, 1964, pp. 137-141). The work showed how a compromised lens epithelium can lead initially to posterior subcapsular cataract, and ultimately to complete lens opacity. The work showed that X-ray affected only proliferating epithelial cells present in the equatorial region of the lens. Using shielding, it was shown that if the germinative region of the lens was protected, very high doses of X-ray could be passed through the lens center without causing any loss of transparency since cells in the lens center do not divide (Pirie A and Flanders PH, AMA Arch Ophthalmol., 1957). Also, young lenses had a much greater susceptibility to X-ray cataract compared to older lenses because of their more rapidly dividing population of epithelial cells (Cogan DG and Donaldson DD, AMA Arch. Ophthalmol., 1951).  One wonders how all of this might have affected the results of the current study, based on what stage of the cell cycle the six different cell lines were in when irradiated.

OK : we agree with the reviewer that a number of further questions can be asked in this study. However in the next comments the reviewer finds that we did not have focus on the major aims of this study :

  • the first citation concerns the localization of the cataract in vivo : obviously, we cannot predict from in vitro cell cultures where the cataract will be located, as far as the reply to this question depends also on the nature of the radiation type (see below)

  • The second citation is related to very high doses of X-ray whose effects cannot be extrapolated to humans but also between the different animal species, because of the great differences in radiosensitivity. Such question is therefore not in the scope of our study.

  • The third citation concerns the age of the lens donor and its link with predisposition to and/or occurrence of cataract. However, from the second citation, it is well said that the dose is a crucial factor for cataractogenesis. Furthermore, significant cohorts of adults (cardiologists and astronauts) show radiation-induced cataracts. Hence, again, data developed in this study are not sufficient or are not relevant to estimate the relative (and separate) contribution of the dose and the age of donor in the cataractogenesis process.

As a whole, the reviewer should be aware that this work represents a preliminary study and only 6 porcine lens cells cannot reply to these important questions. See modified text in conclusions.

This manuscript was difficult for this reviewer to comprehend. It appeared that a considerable amount of false starts and negative data were included in the paper that could have been left out in order to focus more on the main finding of the study. Only a small proportion of the data contained in the 17 figures and one table of the paper are mentioned in the abstract.  

We are aware that there is a huge amount of data. However, first, we deliberately chosen to show a maximal part of what we have done, including our failures since we think that it can be useful for groups of research that aim to establish lens cell lines. Second, a recurrent comment of the reviewer is that figures are not large enough : we try to reach such requirement to make our figures more readable. Lastly, the second reviewer ask us to complete some discussion and did not comment the number of figures.

The incorrect use and non-use of the article "the" is frequent in the manuscript.

OK English has been edited.

Reviewer 2 Report

The manuscript presented by Al-Choboq is well written and covers an important topic, leading to deeper insight into ATM-dependend radiation response. The target, lens, is well known for its sensitivity for radiation-induced changes, so dealing with the mechanism behind radiation induced-pathologies is of high interest.

I have some few issues I would like to have addressed or commented on. 

1. yH2AX foci count: you do suggest that fewer foci might be related to not yH2AX, but still existing foci (p7, line 273). In general I'm in accordance with this. However, since you have nicely presented the different nucleus sizes of the different cell lines, you should take these differences into consideration. Given the physical characteristic of X-ray a larger nucleus size would give a higher number of foci. The same is true for the PFS-1 cell line (p11, line 423 ff). Fibroblast tend to have a larger nucleus. You have not given numbers for this cell line in the manuscript, but you should check for size-corrected numbers of foci in this regard as well.

2. MRE-11-foci in HLEpiC (p8, line 330 / Fig. 5): You estimate that there is almost no change from 4 to 24hrs post irradiation, even higher levels at 24hrs were proposed. Even though as the chosen evaluated time points are common for radiation experiments I think in this case at least one intermediate time point would have been necessary. It is possible that you missed the peak of MRE-11 foci formation in the unobserved 20hrs. I would recommend to extent this experiment for additional time points between 4 and 24 hrs.

3. p-ATM-foci in HLE-B3 vs. HLEpiC (p9, line 357): you state that the number of foci at 10 min is significant lower in the HLEpiC, which is absolutely true. However your data in figure 6C show that at 10 min the maximum of p-ATM-foci formation has not yet been reached. It is known that different cell lines/types tend to differ in the time frame until max of foci formation is reached. This should be taken into consideration. According to your figure at max the HLE-B3 would reach almost 20 foci.

4. pATM-foci in PSF1 vs. CP (p12, line 442): similar effect as in issue nr. 3: the peak of p-ATM foci is obviously closer to 1hr than to 10 min. This needs to be taken into consideration when making a conclusion about radio sensitivity. Again it is known that some cell lines peak later in this matter.

5. p16 doubling: you calculated values according a hypothetical doubling time of 1 or 2 doublings per passage. Do you have the generation time per cell line calculated, respective counted cell numbers per mL at seeding and at passage? It would be helpful to correlate it with real doubling times, since it would help to better compare the results from the single cell lines and check. Since especially CP3 shows really high values after 12 passages it would be good to know if this correlates with a fast doubling time and therefore an "older" cell population at passage 12 

Further issues:

syndromic X-proteins: Maybe I'm not aware of this phrase, but is not totally clear what this includes. Usually X-linked syndromic phenomena are linked to protein mutations/functions/failures coded on the X-chromosome. A short literature search I made left me non the wiser. Please elaborate if this is a phrase you introduced for summarizing the unkown proteins binding to ATM or what exactly is meant by this.

Figure 13: it is unclear to which sets of bars the significance (**) is related.

Typos:

p1, line 31: crownd

p15, line 529: AM>T ? 

So, since mostly the data related to the issues I named are available they just need to be taken into account when interpreting your results. The only experiment that need further experimental input are MRE-11 foci evaluations. Here additional timepoints to omit missing the MRE-11 foci peak can give important insights. However in my opinion, this can also be thoroughly discussed in this manuscript and needs for result interpretation definitely be considered. Nevertheless, it should in any case be taken into account in the already mentioned follow-up studies dealing with these cell lines.

Author Response

Replies to the reviewer 2 ‘s comment

We thank the reviewer’s for his/her comments

The manuscript presented by Al-Choboq is well written and covers an important topic, leading to deeper insight into ATM-dependend radiation response. The target, lens, is well known for its sensitivity for radiation-induced changes, so dealing with the mechanism behind radiation induced-pathologies is of high interest.

I have some few issues I would like to have addressed or commented on. 

  1. yH2AX foci count: you do suggest that fewer foci might be related to not yH2AX, but still existing foci (p7, line 273). In general I'm in accordance with this. However, since you have nicely presented the different nucleus sizes of the different cell lines, you should take these differences into consideration. Given the physical characteristic of X-ray a larger nucleus size would give a higher number of foci. The same is true for the PFS-1 cell line (p11, line 423 ff). Fibroblast tend to have a larger nucleus. You have not given numbers for this cell line in the manuscript, but you should check for size-corrected numbers of foci in this regard as well.
  2. The major basis of the RIANS model is that, if cytoplasmic ATM monomers are sequestrated by X-proteins, the flux of ATM monomers that diffuse in the nucleus decreases. Consequently, there will be less available ATM monomers to phosphorylate the H2AX histone at the DSB sites. Hence, for the same number of DSB physically induced, the number of yH2AX foci decreases. In the different cell lines tested, the average size of the nucleus ranged from 1386 to 730 um2 (i.e : a variation of about 47%) and the number of early yH2AX foci from 80 to 40 (i.e. a variation of about 50%). However, the change of size cannot explain the difference in yH2AX foci. For example, the HLEpic cells elicit the largest nuclei but also one of the lower yH2AX foci value. Similarly, the nuclei of PSF-1 cells are smaller than the CP ones while the CP cell lines showed lower numbers of early yH2AX foci. Furthermore, nuclei of human and porcine fibroblasts were not significantly different while they showed 80 and 50 early yH2AX foci at 10 min post-irradiation, respectively. Lastly, it must also stressed that each species and each cell type may have different pools of available ATM monomers but also different pools of X-proteins, which renders more complex the potential link (if any) between the number of early yH2AX foci and other features specific to each cell type. In our experience, only the RIANS hypothesis is mathematically compatible with our findings (see Granzotto et al. IJROBP, 2016). See modified text line 275 and table 1
  1. MRE-11-foci in HLEpiC (p8, line 330 / Fig. 5): You estimate that there is almost no change from 4 to 24hrs post irradiation, even higher levels at 24hrs were proposed. Even though as the chosen evaluated time points are common for radiation experiments I think in this case at least one intermediate time point would have been necessary. It is possible that you missed the peak of MRE-11 foci formation in the unobserved 20hrs. I would recommend to extent this experiment for additional time points between 4 and 24 hrs.
  2. The reviewing time does not permit us to re-do experiments and add new data. However, there are two major arguments to explain that no peak between 4 and 24 h should be observed. First, at 1 h 50% of recognition and DNA repair is done, at 4 h more than 80% repair is done. If a peak would appear between 4 and 24 h, it should be larger than the 7-12 foci values observed and represents 20% of the DNA damage unrecognized : it is not mathematically possible. Besides, the same conclusions can be reached with yH2AX and pATM. Second, from or collection of cell lines gathering more than 350 cases and covering the large spectrum of human radiosensitivity, we provided evidence in Bodgi et al. 2013 JTB (ref 43)., that all the kinetics of appearance/disappeareance of nuclear foci (including MRE11) obey the so-called Bodgi’s formula that illustrates well the different constraints that formation of nuclear foci must follwo during post-irradiation repair time. Hence, in this study, all the MRE11 foci kinetics obey the Bodgi’s formula and no peak between 4 and 24 h has been still observed. Let’s remind again that yH2AX, pATM and MRE11 data are intimately and kinetically linked and all this group of data is mathematically coherent (see ref 44 and 45). See also modified text in materials and methods.

  1. p-ATM-foci in HLE-B3 vs. HLEpiC (p9, line 357): you state that the number of foci at 10 min is significant lower in the HLEpiC, which is absolutely true. However your data in figure 6C show that at 10 min the maximum of p-ATM-foci formation has not yet been reached. It is known that different cell lines/types tend to differ in the time frame until max of foci formation is reached. This should be taken into consideration. According to your figure at max the HLE-B3 would reach almost 20 foci.
  2. You are fully right. Furthermore , a ratio yH2AX/pATM of 2 should be obtained. However, in practice, the shape of pATM foci in B3 made difficult the foci scoring and we have modified the text accordingly line 360.

  1. pATM-foci in PSF1 vs. CP (p12, line 442): similar effect as in issue nr. 3: the peak of p-ATM foci is obviously closer to 1hr than to 10 min. This needs to be taken into consideration when making a conclusion about radio sensitivity. Again it is known that some cell lines peak later in this matter.
  2. Again you are right. but here the error bars indicate that there is not significant difference between the 10 min and 1 h data in CP cells and therefore, no clear conclusion with regard the maximal value can be done rigorously : sntences are correct. Note that the conclusion about radiosensitivity is related to the fact that the numbers of early yH2AX foci are much lower in the first hour post-irradiation than expected (30 pATM foci instead of the expected 40 at 10 min).

  1. p16 doubling: you calculated values according a hypothetical doubling time of 1 or 2 doublings per passage. Do you have the generation time per cell line calculated, respective counted cell numbers per mL at seeding and at passage? It would be helpful to correlate it with real doubling times, since it would help to better compare the results from the single cell lines and check. Since especially CP3 shows really high values after 12 passages it would be good to know if this correlates with a fast doubling time and therefore an "older" cell population at passage 12
  2. Skin fibroblast and epithelial lens cells show contact inhibition in vitro and therefore their maximal number of cells per cm2 is necessarily limited in Petri dishes and flasks. In our conditions, 1 or 2 doubling passage have been observed. But even if we consider an error of ± 1 passage, it cannot mathematically explain the differences in the pATM crowns data ranging from 20 to 80. The same conclusion is reached with the number of seeded cells whose intercell differences do not reach a factor 4 like the pATM crowns data. Furthermore, it is noteworthy that the nuclei size pf CP cells is not significantly different. Hence, the surface of nuclei to be covered by ATM dimers during the formation of the crowns should not be different between the CP cells. However, some individual differences did appear. Their origin is likely to be related to the X-proteins abundancy that help the initiation of the formation of the pATM crowns. In the frame of the RIANS model, this feature is directly linked to the individual factor and not to technical artefacts. Besides, we worte in the first version that such differences may illustrate the individual predisposition. See modified text line 551.

Further issues:

syndromic X-proteins: Maybe I'm not aware of this phrase, but is not totally clear what this includes. Usually X-linked syndromic phenomena are linked to protein mutations/functions/failures coded on the X-chromosome. A short literature search I made left me non the wiser. Please elaborate if this is a phrase you introduced for summarizing the unkown proteins binding to ATM or what exactly is meant by this.

OK there is some misunderstanding. In the frame of the RIANS model, we called X-proteins, the ATM substrates that are abnormally overexpressed in the RIANS-delayed cells. These proteins were called X because, they are potentially unknown but they have nothing in common with X chromosome. In a case of a well-identified genetic disease, the over-expressed X-protein that causes the sequestration of ATM monomers in the cytoplasm is generally that whose (heterozygous) mutations cause the syndrome : they have been called syndromic X-proteins. However, some tissues may also show naturally some overexpresseed proteins in cytoplasm, independently of any disease. If, they are ATM substrates, they are potential X-proteins. We called them tissue X-proteins. Their action may be added to the syndromic X-proteins in a context of an individual affected by a well-known syndrome whose a major feature is found in a specific tissue. See modified from line 652.

Figure 13: it is unclear to which sets of bars the significance (**) is related.

Ok see modified text in Figure 13 caption

Typos:

p1, line 31: crownd

OK see modified text in abstract

p15, line 529: AM>T ? 

OK see modified text

So, since mostly the data related to the issues I named are available they just need to be taken into account when interpreting your results. The only experiment that need further experimental input are MRE-11 foci evaluations. Here additional timepoints to omit missing the MRE-11 foci peak can give important insights. However in my opinion, this can also be thoroughly discussed in this manuscript and needs for result interpretation definitely be considered. Nevertheless, it should in any case be taken into account in the already mentioned follow-up studies dealing with these cell lines.

We have reached all the requirements of the reviewer.